# Production and Application of Stable Isotope-Labeled Internal Standards for RNA Modification Analysis

**DOI:** 10.3390/genes10010026

**Published:** 2019-01-05

**Authors:** Kayla Borland, Jan Diesend, Taku Ito-Kureha, Vigo Heissmeyer, Christian Hammann, Amy H. Buck, Stylianos Michalakis, Stefanie Kellner

**Affiliations:** 1Department of Chemistry, Ludwig Maximilians University Munich, Butenandtstr. 5-13, 81377 Munich, Germany; kaboch@cup.uni-muenchen.de; 2Department of Life Sciences and Chemistry, Jacobs University Bremen GmbH, Campus Ring 1, 28759 Bremen, Germany; j.diesend@jacobs-university.de (J.D.); c.hammann@jacobs-university.de (C.H.); 3Institute for Immunology at the Biomedical Center, Ludwig-Maximilians-Universität München, 82152 Planegg-Martinsried, Germany; Taku.Kureha@med.uni-muenchen.de (T.I.-K.); vigo.heissmeyer@med.uni-muenchen.de (V.H.); 4Helmholtz Zentrum München, Research Unit Molecular Immune Regulation, Marchioninistr. 25, 81377 Munich, Germany; 5Institute of Immunology & Infection and Centre for Immunity, Infection & Evolution, School of Biological Sciences, University of Edinburgh, Edinburgh EH9 3FL, UK; a.buck@ed.ac.uk; 6Center for Integrated Protein Science Munich CiPSM at the Department of Pharmacy—Center for Drug Research, Ludwig-Maximilians-Universität München, Butenandtstr. 5-13, 81377 Munich, Germany; stylianos.michalakis@cup.uni-muenchen.de

**Keywords:** absolute quantification of RNA modifications, isotope labeling, mass spectrometry, transfer RNA, mRNA

## Abstract

Post-transcriptional RNA modifications have been found to be present in a wide variety of organisms and in different types of RNA. Nucleoside modifications are interesting due to their already known roles in translation fidelity, enzyme recognition, disease progression, and RNA stability. In addition, the abundance of modified nucleosides fluctuates based on growth phase, external stress, or possibly other factors not yet explored. With modifications ever changing, a method to determine absolute quantities for multiple nucleoside modifications is required. Here, we report metabolic isotope labeling to produce isotopically labeled internal standards in bacteria and yeast. These can be used for the quantification of 26 different modified nucleosides. We explain in detail how these internal standards are produced and show their mass spectrometric characterization. We apply our internal standards and quantify the modification content of transfer RNA (tRNA) from bacteria and various eukaryotes. We can show that the origin of the internal standard has no impact on the quantification result. Furthermore, we use our internal standard for the quantification of modified nucleosides in mouse tissue messenger RNA (mRNA), where we find different modification profiles in liver and brain tissue.

## 1. Introduction

In the central dogma of molecular biology, messenger RNA (mRNA) is the molecule linking the genetic code, DNA, with the cellular work force, proteins. mRNA is translated into proteins with the help of transfer RNA (tRNA) and ribosomal RNA (rRNA). In addition to these key players, many other RNA species have been identified such as long non-coding RNAs (lncRNAs), micro RNAs (miRNAs) or even enzymatic RNAs like RNase P [1]. For over half a century, it is known that tRNAs are post-transcriptionally modified [2]. These modified nucleosides and the enzymes involved in their generation are crucial to cell homeostasis. Gene defects of RNA modifying enzymes lead to complex diseases, commonly neurological diseases (for review see [3,4]). The modification status of other RNA types was undefined for a long time. With the development of more sensitive mass spectrometric tools and sequencing methods, the modification status of single RNAs or the whole transcriptome became accessible [5,6,7]. Modification mapping by sequencing is an indirect tool, which uses signals like reverse transcription stalling or stops to detect a modification [7]. Thus, it cannot clearly assign the chemical identity of the detected modification and rigorous validation is necessary [8]. The chemical identity of a modified nucleoside is determined by mass spectrometry. Mass spectrometry relies on the purification of the RNA of interest followed by a complete or partial digestion. Partial digestion with RNases like RNase T1 leads to the formation of short oligonucleotide stretches and the identity and sequence context of a modification can be determined [9]. Here, several micrograms of purified RNA are necessary. Usually such quantities are not available, especially for low abundant RNAs such as miRNAs or defined mRNAs. Here, only nanogram amounts of purified target RNA are available and subjected to complete enzymatic digestion to the nucleoside building block. Nucleoside analysis has the advantage of unparalleled sensitivity and the detection of some modified nucleosides is possible down to amol quantities. Thereby, the modification profile of low RNA amounts can be determined but at the price of the lost sequence context [10]. Another drawback of mass spectrometry is the non-quantitative nature of the technique. The detection efficiency of an analyte depends on many, partly non-controllable, factors in mass spectrometry [11]. These lead to signal fluctuations, and thus a reliable and absolute quantification is not possible. To overcome this obstacle, stable isotope-labeled compounds have been utilized as internal standards. Here, heavy isotopes like carbon-13, nitrogen-15 or oxygen-18 are used to produce the analyte of interest. These heavy isotopologues of the target analyte have the same physico-chemical properties as the analyte and thus they can be used to correct for mass spectrometric detection fluctuations. The use of deuterium (D) is also possible, but here some physico-chemical properties vary (e.g., retention time) and thus D-labeled isotopomers are not ideal as internal standards [12]. The production of a heavy labeled compound can be done either synthetically [13] or by metabolic labeling [14]. In many fields, metabolic isotope labeling for production of stable isotope-labeled internal standards (SILIS) is established [15,16,17]. For the analysis of modified nucleosides of RNA, Kellner and Helm established metabolic SILIS production in 2014 [14]. Here, the generation of 11 modified nucleosides, mainly from tRNA, was presented. Since then, the technique has found wide distribution in the field of nucleic acid modification analysis [5,18,19,20]. With the use of SILIS, mass spectrometry is now a sensitive, selective and accurate tool for modern analysis and quantification of RNA modification profiles. While the key concept of the method presented in 2014 was highly valuable to the field, it had several disadvantages. With just 11 modified nucleosides, it was not possible to assess the complete modification landscape of e.g., eukaryotic tRNAs. Major RNA modifications like hypermodified uridines (position 34) or hypermodified adenosines (position 37) could not be quantified with the method. In addition, the original method used total RNA for SILIS production. The modified nucleosides from tRNA were diluted with excess canonical nucleosides from the scarcely modified rRNA. Thus, a high amount of canonical nucleosides was added to the sample through the SILIS, which interfered with ultraviolet (UV) quantification of the canonical nucleosides in the sample. 

Here, we present recent developments in the production of SILIS for accurate quantification of modified nucleosides, which overcome these disadvantages. We use bacterial RNA from *Escherichia coli* and eukaryotic RNA from *Saccharomyces cerevisiae*, which allows the quantification of up to 26 modified nucleosides. In addition, we only use purified large RNA and tRNA for production of our new SILISs. We present a simple overview to determine which SILIS is best for the quantification of a desired modified nucleoside or RNA type such as tRNA or mRNA. Although the SILISs are produced in simple microorganisms, it can be applied to samples from all domains of life. To demonstrate its strength, we study and compare the modification profile of tRNA from *E. coli* and *Pseudomonas aeruginosa*. In addition, we compare the modification profiles of tRNA from various eukaryotes, including fungi, amoebozoa, worms, mouse tissue and human cell culture. Finally, we demonstrate the use of a tRNA derived SILIS for the quantification of modified nucleosides in purified mRNA samples from mouse brain and liver tissue.

## 2. Materials and Methods 

### 2.1. Chemicals and Reagents

All salts were obtained from Sigma Aldrich (Munich, Germany) at molecular biology grade unless stated otherwise. Isotopically labeled compounds: ^15^N-NH_4_Cl (≥98% atom, Sigma-Aldrich). ^13^C_6_-glucose (≥99% atom, Sigma-Aldrich) and l-methionine-methyl-D_3_ (98 atom % D, Sigma-Aldrich). All solutions and buffers were made with water from a Millipore device (Milli-Q, Merck, Kenilworth, NJ, USA). 

Appendix A shows all synthetic standards of modified nucleosides and their respective vendors.

### 2.2. Labeling of *Escherichia coli*

A single colony of *E. coli* BW25113 grown on a lysogeny broth (LB) agar plate was selected for inoculation of 5 mL M9 media liquid pre-culture. 

Minimal M9 media was used with indicated isotopes. The M9 was prepared by mixing a 10× M9 salt stock solution to a 1× final concentration with *^13^C_6_-glucose* (final concentration 0.4%), MgCl_2_ (final concentration 2 mM), Na_2_SO_4_ (final concentration 2 mM) and CaCl_2_ (final concentration 0.1 mM).

10× M9 stock solution was made with Na_2_HPO_4_, KH_2_PO_4_, NaCl and ^15^N-NH_4_Cl at the respective final concentrations (482, 220, 43, 187 mM). All solutions were sterile filtered and stored at room temperature until use.

The pre-culture was allowed to grow overnight at 37 °C with 250 rpm shaking and was then used to inoculate a 200 mL culture. The 200 mL culture was grown overnight and was harvested when it reached an OD_600_ (optical density) of 2.2 in early stationary phase.

### 2.3. Labeling of *Saccharomyces cerevisiae*

A single colony of *S. cerevisiae* BY4741 grown on a yeast extract peptone dextrose (YPD) plate agar plate was selected for inoculated of 5 mL supplemented Silantes ^13^C media for liquid culture. Silantes ^13^C media was supplemented with *^13^C_6_-glucose* (final concentration 1%) and CD_3_ methionine (final concentration 0.1 g/L). Liquid pre-culture was incubated overnight at 30 °C at 250 rpm. The pre-culture was used to inoculate two 100mL cultures. Cultures were grown overnight to late exponential/early stationary with an OD_600_ of 3.5.

### 2.4. Isolation of Total RNA for Stable Isotope-Labeled Internal Standards (SILIS) Production

Bacteria or yeast cultures were centrifuged at 1200× *g* for 5 min to pellet the cells. The resulting cell pellet was resuspended in 1 mL TRI-Reagent^®^ (Sigma-Aldrich) per 5–10 mL of culture. For every 1 mL of the TRI-Reagent^®^ used, 200 µL of chloroform (≥99% purity, Roth, Karlsruhe, Germany) was added to the solution and the mixture was vortexed. In case of *S. cerevisiae*, cell lysis was aided by addition of acid-washed glass beads as previously described [21]. The biphasic solution rested at room temperature for 5 min and was centrifuged at room temperature for 10 min at 12,000× *g*. The resulting clear upper phase was precipitated with an equal volume of isopropanol (Roth) After incubation at −20 °C overnight, the precipitated total RNA was pelleted by centrifugation for 20 min at 12,000× *g* at 4 °C. The RNA pellet was washed two times with 100–200 µL of 70% EtOH and dissolved in 30 µL water.

### 2.5. Sample Preparation and RNA Isolation

#### 2.5.1. *Dictyostelium Discoideum*

The cultivation of the *Dictyostelium discoideum* AX2 wild-type strain was carried out at 22 °C in constant light. Cells were grown axenically in HL5+ medium (Formedium, Norfolk, UK) in shaking culture. 2 × 10^7^ cells were collected via centrifugation at room temperature and washed once with 1× Soerensen buffer (2 mM Na_2_HPO_4_, 15 mM KH_2_PO_4_, adjusted with H_3_PO_4_ to pH 6.0). Cells were lysed with 1 mL Trizol^®^ reagent with 20 mM EDTA (ethylenediaminetetraacetic acid) and incubated for 5 min at room temperature. Isolation of total RNA was performed according to manufacturer’s protocol and subsequent size exclusion chromatography (SEC) purification as previously described [22]. 

#### 2.5.2. *Caenorhabditis Elegans*

*Caenorhabditis elegans* cultures were grown to the gravid adult stage and harvested and flash frozen as in [23]. Total RNA was isolated with TRI-Reagent^®^ as described in Section 2.4.

#### 2.5.3. Cell Culture

HEK 293T cells were cultured in Dulbecco’s modified Eagle medium (DMEM) (Sigma Aldrich) with 10% fetal bovine serum (FBS) (Sigma) at 37 °C with 10% CO_2_. Cell were seeded at 20% conflunce and allowed to grow to 80% confluence prior to harvesting. Cells were harvested directly in cell culture flasks using 1 mL TRI-Reagent^®^ and total RNA was isolated as previously reported [22].

#### 2.5.4. Mouse Brain and Liver

Female C57BL/6J mice (six months old) were used. The mice were anesthetized with isoflurane and subsequently euthanized by cervical dislocation. Mouse organs were isolated and frozen in liquid nitrogen. The organs were kept at −80 °C until RNA extraction was performed. For RNA isolation, 50–100 mg of tissue were sliced on ice and homogenized in 1 mL TRI-Reagent^®^ using a Qiagen TissueRuptor II and disposable probes (Hilden, Germany). Subsequent isolation of total RNA was performed as previously described.

### 2.6. Purification of Ribosomal RNA and Transfer RNA by Size Exclusion Chromatography

For purification of tRNA from total RNA SEC [24] was used on an Agilent 1100 high-performance liquid chromatography (HPLC) system (Degasser, G1279A; Quat Pump, G1311A; ALS, G1313A; COLCOM, G1316A; VWD, G1314A; Analyt FC, G1364C) with an AdvanceBio column, 300 Å pore size, 2.7 µm particle size, 7.8 × 300 mm (Agilent, Waldbronn, Germany). RNA was eluted with an isocratic gradient with a flow rate of 1mL/min of 0.1 M ammonium acetate (≥98%). Eluting RNA was detected at 254 nm with a diode array detector. Under these conditions tRNA elutes at a retention time between 7 and 8 min. The 1 mL tRNA fraction was collected and evaporated (GeneVac, EZ-2 PLUS, Ipswich, UK) to a volume of ~100 µL before precipitation by a standard ammonium acetate is the precipitation at −20 °C overnight. The tRNA was pelleted by centrifugation (12,000× *g*, 20 min, 4 °C) and washed with 70% ethanol and resuspended in 30 µL water.

### 2.7. Messenger RNA Purification

From isolated total RNA, a 100 μg aliquot was used for mRNA purification with a Dynabeads mRNA purification kit (catalog number 61006) purchased from Thermo Fisher Scientific (Waltham, MA, USA). Manufacturer instructions were followed exactly. Resulting RNA was further treated with Ribominus Transcriptome Kit (catalog number K155001, Thermo Fisher Scientific) to remove any residual rRNA. Again manufacturer instructions were followed without deviation. Resulting RNA was treated again with a Dynabeads mRNA purification kit. To verify the depletion of rRNA and tRNA, the mRNA samples were analyzed with the Bionalyzer RNA 6000 Nano assay on an Agilent 2100 Bioanalyzer System. 

### 2.8. Enzymatic Digestion of RNA

10 µg to 100 ng portions of RNA were digested to single nucleosides with Alkaline Phosphatase (0.2 U, Sigma-Aldrich, St. Louis, MO, USA), Phosphodiesterase I (0.02 U, VWR, Radnor, PA, USA) and Benzonase (0.2 U) in Tris (pH 8, 5 mM) and MgCl_2_ (1 mM) containing buffer. Furthermore, tetrahydrouridine (THU, 0.5 µg from Merck), butylated hydroxytoluene (BHT, 1 µM) and Pentostatin (0.1 µg) were added to protect modifications [25]. The mixture was incubated with the RNA for two hours at 37 °C. Afterwards samples were filtered through 96 well filter plates (AcroPrep^TM^ Advance 350 10K Omega^TM^, PALL Corporation, Port Washington, NY, USA) or single tubes (VWR, 10 kDa MWCO) at 4 °C for longer than 10 min at 3000× *g* to remove digestive enzymes. 

### 2.9. High-Resolution Mass Spectrometry

The ribonucleosides were separated using a Dionex Ultimate 3000 HPLC system on an RP-18 column (Synergi, 2.5 µm Fusion-RP C18 100 Å, 100 × 2 mm; Phenomenex^®^, Torrance, CA, USA). Mobile phase A was 10 mM ammonium acetate and mobile phase B was 80% acetonitrile. The gradient began with 0% B and increased to 20% B over 10 min and to 80% by 12 min. After a 4 min hold at 80% B, the column returned to 100% A over 1 min. The column was re-equilibrated with 100% A for 8 min. The flow rate was 0.2 mL/min with a column temperature of 30 °C. For untargeted analysis, the slower flow rate resulted in better peak separation and thus mass spectra generation for individual compounds.

High-resolution mass spectra of precursor and product ions were recorded by a Thermo Finnigan LTQ Orbitrap XL operated in positive ionization mode with a capillary voltage of 20 V and temperature of 275 °C. Sheath gas flow was set to 5, and auxiliary gas was set to 35. 

### 2.10. Dilution of RNA Digests for 10× SILIS Generation

Serial dilutions of each SILIS type were prepared for analysis to determine the proper dilution A dilution with peaks at least 10-fold above the lower limit of quantification (LLOQ) for the lowest abundant modified nucleosides was considered ideal.

### 2.11. Liquid Chromatography–Mass Spectrometry (LC–MS/MS) Analysis

For quantification a 1290 Infinity II (Agilent Technologies, Waldbronn, Germany) equipped with a diode array detector (DAD) combined with a G6470A Triple Quad system (Agilent Technologies) and electro-spray ionization mass spectrometry (ESI-MS, Agilent Jetstream; Agilent Technologies) was used. Operating parameters were as follows: positive ion mode, skimmer voltage 15 V, Cell Accelerator Voltage 5 V, N_2_ gas temperature 230 °C and N_2_ gas flow 6 L/min, sheath gas (N_2_) temperature 400 °C with a flow of 12 L/min, Capillary Voltage of 2500 V, Nozzle Voltage of 0 V and the Nebulizer at 40 psi. The instrument was operated in dynamic multiple reaction monitoring (MRM) mode with the method listed in Appendix A and the individual mass transitions for the nucleosides are given in Appendix A. The MRM approach allowed shorter runs as separation of individual compounds was not crucial for analysis. The mobile phases were: A as 5 mM NH^4^OAc (≥99%, HiPerSolv CHROMANORM^®^, VWR) aqueous buffer, brought to pH = 5.6 with glacial acetic acid (≥99%, HiPerSolv CHROMANORM^®^, VWR) and B as pure acetonitrile (Roth, liquid chromatography–mass spectrometry (LC–MS) grade, purity ≥ 99.95). A Synergi Fusion-RP column (Phenomenex^®^, Torrance, CA, USA; Synergi^®^ 2.5 µm Fusion-RP 100Å, 150 × 2.0 mm) at 35 °C and a flow rate of 0.35 mL/min was used. The gradient began with 100% A for one minute, increased to 10% B by 5 min, and to 40% B by 7 min. The column was flushed with 40% B for 1 min and returned to starting conditions to 100% A by 8.5 min followed by re-equilibration at 100% A for 2.5 additional minutes. 

## 3. Results

### 3.1. Generation and Characterization of SILIS

Our laboratory analyzes samples from bacteria and various eukaryotes on a regular basis. While many modified nucleosides overlap in both domains of life, some are unique to one of them. For example, 2-methyladenosine (m^2^A) is only found in bacteria, while 1-methyladenosine (m^1^A) and 3-methylcytidine (m^3^C) are common in eukaryotes but not in bacteria. For absolute quantification, isotope dilution mass spectrometry is the method of choice and thus stable isotope labeled internal standards, SILIS, are needed. In 2014 the production of 11 modified nucleosides by metabolic labeling was shown. Important modified nucleosides e.g., the hypermodified uridines and adenosines (e.g., mcm^5^s^2^U or i^6^A) were not reported. With the goal to expand the number of metabolically produced stable isotope labeled nucleosides, we decided to prepare SILIS in the bacterium *E. coli* and in addition, in the eukaryote *S. cerevisiae*. At this point, it was necessary to decide which labeling strategy should be used. For standard quantification of unlabeled samples, a simple labeling scheme (e.g., in *E. coli* only carbon-13) is possible and most economic. We do not recommend nitrogen-15 labeling alone in *E. coli* for SILIS production. The resulting heavy uridine nucleosides would be only 2 Dalton heavier compared to the unlabeled nucleosides, which is not sufficient due to the natural abundance of ^13^C in the natural nucleosides. Our laboratory is specialized in the analysis of isotopically labeled nucleosides from e.g., nucleic acid isotope labeling coupled mass spectrometry (NAIL-MS) pulse-chase experiments [21,26]. These samples already contain isotopically labeled nucleosides and are thus up to 8 Dalton heavier than unlabeled nucleosides. In our context, the SILIS must be even heavier than the samples. Therefore, we decided to use a simultaneous nitrogen-15 and carbon-13 labeling in *E. coli* and a combined carbon-13 and CD_3_-methionine labeling in *S. cerevisiae*. Deuterium labeling of a single methyl-group, as introduced by CD_3_-methionine, does not lead to detectable changes in the physicochemical properties of the methylated nucleosides and thus the labeling strategy is suitable for SILIS production [13]. 

With the goal to produce enough SILIS to last several months or even years, *E. coli* was cultured in 200 mL of carbon-13 and nitrogen-15 containing M9 medium. *S. cerevisiae* was grown in two separate 100 mL carbon-13 full medium flasks (Silantes, Munich, Germany) supplemented with ^13^C_6_-glucose and CD_3_ methionine (final concentration 0.1 g/L). Both organisms were grown to late exponential phase and harvested by centrifugation. The total RNA was isolated using TRI^®^ reagent. In our previous work [14], the total RNA was used for generation of the SILIS. Thus, an unnecessary excess of canonical nucleosides was introduced into the sample, which complicated the quantification of canonical nucleosides. Therefore, we decided to fractionate the total RNA into the long RNAs and the heavily modified tRNA fraction by size exclusion chromatography [24] to reduce the amount of canonical nucleosides in the later tRNA SILIS. An aliquot of the purified RNA was enzymatically digested to the nucleoside building block. The digest was immediately injected on a Thermo Orbitrap high resolution mass spectrometer for characterization of the produced nucleoside isotopomers. The principle of SILIS generation is shown in Figure 1. 

The mass spectra of the heavy isotope labeled RNA digests showed a high labeling efficiency for both organisms. We used synthetic standards for retention time comparison and could thus assign all extracted mass spectra to their corresponding modified nucleoside. In Figure 2 (left), the chemical structure, mass spectra and labeled positions are shown for the canonical nucleoside adenosine. In yeast, a ^13^C_10_-labeled adenosine is formed and thus the mass is increased by an m/z of 10. In *E. coli*, we see a ^13^C_10_-^15^N_5_-labeled adenosine and a mass increase of +15. In both RNA digests, the abundance of unlabeled adenosine is negligible. In Figure 2 (right), the results for 6-methyladenosine (m^6^A) are shown. In yeast, we see the formation of the main product, a ^13^C_10_-CD_3_-labeled m^6^A (m/z 295.17). We also observe some m/z 293.17, which corresponds to a ^13^C_11_-labeled m^6^A. The CD_3_-label is introduced by supplementing CD_3_-methionine to the media. Since we used a ^13^C full media, some ^13^CH_3_-methionine is also available to the cells and, thus, we also find ^13^CH_3_-methylated nucleosides in the yeast RNA. For *E. coli*, we see a distinct labeling of ^13^C_11_-^15^N_5_-labeled m^6^A (+16). Appendix A shows the mass spectra of various other nucleosides from the produced labeled RNA digests.

After mass spectrometric characterization, we serially diluted the isotopically labeled RNA digests and analyzed them on a sensitive triple quadrupole instrument. For highest sensitivity, we used the multiple reaction monitoring function of our Agilent software. (For instruments from other suppliers this function is also referred to as single reaction monitoring, SRM) The key parameters for the *E. coli* and *S. cerevisiae* heavy isotope labeled nucleosides are given in Appendix A. We analyzed the dilutions of the digests with their respective parameters and noted the peak intensity of all modified nucleosides. For SILIS preparation, we decided to use a dilution which produces peaks at least 10-fold above the LLOQ of the lowest abundant modified nucleoside. The LLOQ is a unique parameter for each modified nucleoside in the used analytical setup and is determined by analysis of serial dilutions of synthetic standards. From this calibration curve (Appendix A), the LLOQ (Appendix A) is found at the concentration/injected amount where the peak height of the analyte is 10-fold higher than the surrounding noise (S/N > 10).

After determination of the final SILIS concentration, we multiplied the concentration by 10 as we have positive experience with using a 10× SILIS. As an example: For yeast we used 10 µg labeled tRNA and digested the RNA in 30 µL digestion buffer. This digest is diluted 20-fold for production of 600 µL of a 10× yeast SILIS, which is sufficient for analysis of 200–300 samples. In addition, we add a UV-detectable amount of theophylline to our 10× SILIS. Theophyllin absorbs at 260 nm and elutes after adenosine. The theophylline peak area is used to control the correct mixing of SILIS with sample (theophylline acts as an external standard). 

#### Guideline for SILIS Generation

By high-resolution MS characterization, we found a unique signature of modified nucleosides in each of the four produced SILISs (yeast tRNA and largeRNA SILIS and bacteria tRNA and large RNA SILIS). The results are summarized in the Venn diagram of Figure 3 and Table 1. With this information, it is now possible to choose the right organism and RNA type for production of the modified nucleoside of interest. 9 modified nucleosides, such as pseudouridine (Ψ), inosine (I), 7-methylguanosine (m^7^G) and 6-methyladenosine (m^6^A), are found in both RNA types of both organisms (center of Venn diagram, Figure 3) and thus the scientist can decide which organism to use for SILIS generation of these nucleosides.

The modified nucleosides, 4-thiouridine (s^4^U), 2-thiocytidine (s^2^C) and 2-methyladenosine (m^2^A) are unique for *E. coli* tRNA and thus *E. coli* has to be used for SILIS production. While 2-methylguanosine (m^2^G) and 5-methylcytidine (m^5^C) are present in the large RNA of *E. coli*, they are more commonly found in the tRNA of yeast. Here, the production is possible in *E. coli*, but production in yeast would lead to higher yields of SILIS. 5-methyluridine (m^5^U, sometimes also ribothymidine rT), dihydrouridine (D), 6-threonyladenosine (t^6^A) and 6-isopentenyladenosine (i^6^A) are unique tRNA modifications and thus only found in tRNA from both organisms. As a eukaryotic organism, *S. cerevisiae* has several unique RNA modifications. For example 3-methylcytidine (m^3^C), 1-methyladenosine (m^1^A), 2,2,-dimethylguanosine (m^22^G) and 4-acetylcytidine (ac^4^C) are uniquely found in the yeast SILIS and thus must be produced in yeast. Eukaryotes also carry hypermodified uridine derivatives at position 34 of their tRNAs. These wobble uridines are also found in the yeast tRNA SILIS. At this stage we already want to stress that wobble uridines are prone to decomposition due to the conditions of enzymatic digestion. Therefore, the samples and also the yeast tRNA SILIS need to be prepared fresh for absolute quantification of these modified nucleosides. Another hypermodified nucleoside is queuosine (Q). We could clearly find Q in our yeast tRNA SILIS. Q is one of the most complex modified nucleosides, and its synthesis is highly complicated [27]. Therefore, Q is not available as a stable isotope labeled standard or even as an unlabeled standard, which highlights the strength of metabolic labeling for SILIS production. 

Table 1 gives an overview of all modified nucleosides, we identified in our SILIS in alphabetical order. This list was generated from all the high-resolution mass spectra we acquired. In principle, it is also possible to predict the mass transition of an unlisted modified nucleoside and search for it in the respective SILIS. The arising signals should be referenced with the unlabeled synthetic standard for verification. With this method, even lower abundant modified nucleosides are identified in the SILIS and absolute quantification of samples becomes possible.

### 3.2. Absolute Quantification with SILIS

With our 10× SILISs in hand, we decided to use them for quantification of samples of various origins. We used the Venn diagram (Figure 3) to decide which SILIS should be used for which sample type/organism. In general, we digest samples in 20–30 µL and after filtration we mix 18 µL of sample with 2 µL of the respective 10× SILIS and inject 10 µL of the mix onto the LC–MS/MS. For calibration we prepare serial dilutions of the synthetic standards and add 10% SILIS to each solution before injecting 10 µL. A detailed description is given in the Materials and Methods section. After analysis of the calibration solutions, we normalize the signal of the synthetic standard to the signal of the respective internal standard and receive the calibration curve. With the calibration curve, we can then determine the quantities of modified nucleosides in the sample. The quantities of modified nucleosides are normalized against the quantity of one or all canonical nucleoside(s), e.g., guanosine (G) is commonly used [28]. With SILIS quantification from metabolic labeling, we introduce an additional challenge to the quantification of canonical nucleosides. The SILIS contains both modified and canonical nucleosides. Especially total RNA and rRNA SILIS contain a huge excess of canonical nucleosides and these overlap with the sample canonical signal during UV detection. Simple UV detection does not allow the discrimination of the canonicals from SILIS and from the sample. Thus, the amount of sample canonicals is difficult to quantify by UV detection.

To overcome this problem, we use the mass spectrometer to detect and distinguish the canonical nucleosides. With MS detection, we can clearly distinguish SILIS canonicals from sample canonicals due to their different mass transitions. To avoid saturation of the mass spectrometer by the highly abundant canonical nucleosides, we use non-optimal ionization and collision parameters (e.g., elevated fragmentor voltage and high collision energy). Thus, we can clearly quantify the abundance of sample canonicals and use them for normalization of the less abundant modified nucleosides. The quantity of each modified nucleoside per G is then plotted in %. A detailed description of the calculations and normalization steps is given in the Appendix A. 

#### 3.2.1. The Origin of the SILIS Has No Impact on the Quantification Result

In our first experiment, we wanted to demonstrate that the origin of the SILIS used has no impact on the quantification result. For this purpose, we cultured HEK 293T cells and purified the total tRNA [22]. After digestion we split the sample in two aliquots and analyzed one using the *E. coli* tRNA SILIS and the other using the yeast tRNA SILIS. As shown in Figure 4a, we see comparable quantities of modified nucleosides independently of the chosen SILIS. Only Cm shows a 10% difference and i^6^A around 32% difference in dependence of the chosen SILIS. The low abundance of i^6^A in the sample (slightly above LLOQ) is causative for the high difference. The difference for Cm is currently under investigation. Thus we recommend caution upon absolute quantification of Cm. For all other modified nucleosides found in both SILISs, like dihydrouridine (D), 1-methylguanosine (m^1^G), 7-methylguanosine (m^7^G) or 5-methyluridine (m^5^U), we see identical quantification results. For those modified nucleosides that are absent in the *E. coli* tRNA SILIS, no normalization against the stable isotope labeled modified nucleoside was possible and the quantification is less robust (Figure 4b). Here, we used external calibration for quantification of m^1^A, m^22^G, mcm^5^s^2^U and ac^4^C. While m^22^G and mcm^5^s^2^U seem to be robust for correct external calibration, the results for ac^4^C indicate that external calibration is not sufficient and that internal standards are necessary for correct quantification for these modified nucleosides. For four modified nucleosides, we used a chemically similar modified nucleoside as the internal standard in the *E. coli* SILIS. For m^2^G we used the SILIS signal of m^1^G, for Um quantification the SILIS signal for Cm and for m^5^C and m^3^C we used the SILIS signal of m^5^U. In most of these cases, the quantification result is significantly different from the yeast SILIS quantification. Using a chemically similar compound is sometimes sufficient for reliable quantification. However, such an internal standard is unable to normalize the matrix effects and pH effects and thus the quantification results are perturbed as in the case for m^3^C. This experiment highlights the importance of choosing the right SILIS for the analyzed sample type. Our data demonstrates that the origin of the SILIS is mostly without consequence as long as it contains the modified nucleoside of interest. All data can be found in Appendix A.

#### 3.2.2. Modified Nucleosides in *Pseudomonas Aeruginosa* Total tRNA

Bacteria contain a vast variety of modified nucleosides and tRNA modifications are especially getting more attention from microbiologists and infectiologists. Only recently, the tRNA modification status of the pathogen *P. aeruginosa* (PA) was determined [29]. Unfortunately, no absolute quantification of modified nucleosides was possible in this study. Therefore, we decided to grow *P. aeruginosa* PA14 in minimal M9 medium and compare the quantity of modified nucleosides with tRNA from LB and M9 cultured *E. coli* (in stationary growth phase only). In general, we found the same modified nucleosides in *E. coli* and PA tRNA (Figure 5a). For the modified nucleosides m^1^G, m^2^A, m^2^G, s^2^C, m^5^U, m^6^A, m^7^G, Cm and I, we found comparable quantities in both bacterial strains. However, we observed a lower abundance of s^4^U (enzyme: thiI), Gm (TrmH) and Um (TrmJ) in *P. aeruginosa*. Gm, in bacterial tRNA, is connected to inhibition of TLR7, which results in a suppression of the hosts immune system [30]. Our data suggests, that PA tRNA might be more immune-stimulatory compared to *E. coli* tRNA due to the lower abundance of Gm. TrmJ and thus Um on the other hand was shown to be important for oxidative stress survival in PA [29]. At this point, we cannot predict the impact of these low abundant tRNA modifications in PA, but we are confident that future studies will address these findings. Overall, bacteria show a very low modification density in their tRNAs. The absolute quantities are given in Appendix A.

#### 3.2.3. Modified Nucleosides in Total tRNA from Various Eukaryotes

In a next step, we applied our yeast tRNA SILIS to assess the modification profile of tRNAs from increasingly complex eukaryotes. We analyzed *S. cerevisiae*, *D. discoideum*, *C. elegans*, the human cell culture strain HEK 293T and tissue samples from adult mice (here liver and brain). The results are shown in Figure 5b,c and are listed in Appendix A. In comparison to the modification profile of *E. coli* and PA in Figure 5a, we see a higher variety and abundance of modified nucleosides in the eukaryotic samples. This is in accordance with the literature [31]. For the fungus *S. cerevisiae* and the amoebozoa *D. discoideum*, we only observe minor differences in the total abundance of modified nucleosides. One of the most interesting organisms in our analysis is *C. elegans*. *C. elegans* has a similar chemical variety of modified nucleosides as the other eukaryotes, but the abundance is lower compared to the other eukaryotes. The only exception are the ribose methylated nucleosides Um, Gm and Cm, which are of comparable abundance or even increased abundance in the worms (Am). From our data, many hypotheses are possible. For example, the tRNA maturation and modification might be less efficient in the worm and thus more tRNAs remain as unmodified transcripts. It is also possible, that worms contain large quantities of an unknown transcript of 60–100 nucleotide lengths, which is heavily modified with ribose methylation. We are currently investigating why worms display such a bizarre modification profile. Our analysis of the human cell culture samples and the mouse tissue samples revealed an overall high similarity between the sample types. From our data, we observe the previously reported lower abundance of m^2^G, m^1^G and m^22^G in mouse brain compared to mouse liver samples. In addition, we can confirm the comparable abundance of m^1^A and t^6^A in the tRNAs from these tissues [13]. Furthermore, we see a reduced abundance of Gm and Um in the mouse brain tissue compared to liver. 

Samples from eukaryotic organisms, especially cell culture samples, can be contaminated with bacteria. Sometimes it is difficult to detect such contaminations. Here, tRNA modification analysis is a useful tool to determine the contamination status of a sample. Bacteria contain several unique tRNA modifications such as s^4^U, s^2^C or m^2^A. Any of these modified nucleosides can be used as an indicator for bacterial contamination in a eukaryotic sample. However, m^2^A is the most simple of all because of its (a) high abundance in bacterial tRNAs, (b) high detection efficiency and (c) similar retention time to m^6^A. m^6^A is a common eukaryotic modification, which is part of most RNA modification analyses. In case of bacterial contamination, a second peak right before m^6^A appears in the chromatogram of a sample. This second peak is m^2^A and clearly indicates the presence of bacteria. We have had several cases in the past, where we analyzed tRNA from inconspicuous cell cultures and we observed a more or less pronounced peak of m^2^A (see Appendix A). Only later, testing for mycobacteria revealed the actual contamination of the cell culture and the experiment was discarded. Therefore, we recommend careful analysis of m^6^A in eukaryotic samples in order to assess the potential presence of bacterial contaminations in form of m^2^A.

#### 3.2.4. The Abundance of Modified Nucleosides in Messenger RNA Is Different among Tissues

Messenger RNA has become one of the most intensively studied RNA types since the discovery of its dynamic modification status. It was found that m^6^A is incorporated into mRNA by RNA modifying enzymes (RNA writers), where it regulates the binding of RNA reader proteins. In addition, m^6^A and its ribose-methylated form m^6^Am can be removed enzymatically from the mRNA by RNA erasers [32,33,34]. The dynamics of the m^6^A modification in mRNA prompted the start of the field of epitranscriptomics. In addition to m^6^A, other modified nucleosides have been found in mRNA. Some of these are of very low abundance, such as m^5^C, m^3^C and m^1^A. From a mass spectrometrist’s perspective, one of the main challenges of mRNA modification profiling is the availability of a truly pure sample. The Lyko and Helm lab [35] recently demonstrated the impact of the currently used purification protocol on the quantification result and presented evidence for the constant contamination with rRNA. We share their conclusion, that true purification of mRNA is currently not possible and that modified nucleosides found in low abundances are most likely from contaminating RNAs and not from mRNA. The purpose of our next analysis was to demonstrate the difficulties and challenges in mRNA modification analysis. In accordance with the protocol suggested by Lyko and Helm, we have purified total mRNA from mouse liver and mouse brain using first a Poly-A enrichment step, followed by rRNA removal and a second Poly-A enrichment. The quality of the eluted RNA was checked by chip gel electrophoresis. As shown in Figure 6a, we see a size distribution of approximately 500 to 7000 nucleotides, which is to be expected for mRNA. We see no distinct peaks for tRNA or rRNA in the purified RNA. However, the amount of isolated mRNA is very low, and we cannot completely exclude contamination by remnant tRNA or rRNA hidden underneath the mRNA population. We digested the purified mRNA sample and quantified the abundance of modified nucleosides using the yeast tRNA SILIS. Figure 6b shows the results. mRNA is polyadenylated and the length of the Poly-A tail is highly variable and might fluctuate from organ to organ. Thus, we decided against the normalization against A and normalized against G. The most abundant modified nucleoside in both samples is m^6^A. Interestingly, the abundance of m^6^A is significantly higher in mRNA from brain (0.32% m^6^A per G) compared to liver (0.19% m^6^A per G, p-value: 0.00005). m^66^A (also referred to as m^6^_2_A) and m^6^Am were not quantifiable due to their absence in the used dilution of yeast SILIS. The cap modification m^7^G is found in comparable amounts in both tissues. Other modified nucleosides are far less abundant in our isolated mRNA samples and potentially from tRNA or rRNA contamination. Mass spectrometry is a highly sensitive tool, and thus even sub-stoichiometric quantities can be assessed. However, in the case of mRNA preparations, we cannot exclude which origin small quantities of modified nucleosides such as Gm, Cm, Am and Um have. It is quite possible that these signals arise from low amounts of contaminating rRNA. Thus, we recommend caution upon biological data interpretation as many of the quantified nucleosides might be false positive results. Orthogonal techniques, such as sequencing (for review see [36]) must be employed to verify such findings. The only modified nucleoside we want to highlight in addition to m^6^A is m^3^C. This modification was recently described as an mRNA modification in mouse liver and a human cell line, incorporated by METTL8 [37]. The authors of this study found very low abundances of m^3^C (~30-fold less m^3^C than m^6^A) in their cell culture samples. We find a rather high amount of m^3^C in our mouse liver mRNA (5-fold less m^3^C than m^6^A), while in the mouse brain mRNA the abundance is drastically lower (60-fold less m^3^C than m^6^A). The biological significance and correctness needs validation by tools such as the recently presented AlkAnilin-seq [38]. A sequence-specific localization of m^3^C in large RNAs is crucial to understand the origin of the modification. We and others showed that position 3 of cytidine is prone to direct methylation and thus sub-stoichiometric signals of m^3^C might hint towards a chemical methylation and not an enzymatic one [26].

## 4. Discussion

Internal standards are crucial for the correct and robust quantification of modified nucleosides. We have presented the preparation of such standards, also referred to as SILIS, in two microorganisms and characterized the heavy labeled modified nucleosides. In summary, 26 modified nucleosides are easily accessible by our approach, including hypermodified nucleosides. In addition, we present a method which allows the quantification of canonical nucleosides for normalization by mass spectrometry. The key here is the use of non-optimal MS parameters to avoid saturation of the mass spectrometer. By this approach, the canonical nucleosides from the sample can be quantified even in the presence of the canonical nucleosides introduced through the SILIS. From the provided overview, it becomes clear that yeast tRNA is the ideal source for production of SILIS. However, the growth medium for yeast is more expensive than for bacteria (1L costs ~2000 €/yeast and ~400 €/bacteria). In bacteria, 16 eukaryotic modified nucleosides can be produced, however, most only in low abundance (e.g., m^5^C and m^2^G). Nevertheless, bacterial SILIS production might be an economic choice for the analysis of eukaryotic samples. We could show that the origin of the SILIS is of no consequence to the result of quantification as long as the modified nucleoside of interest is actually present in the used SILIS. Overall, we can now perform absolute quantification of up to 26 modified nucleosides, which is 15 more than we previously reported. Among these modified nucleosides, crucial analytes such as the hypermodified uridines and queuosine are found.

We have applied our produced SILISs to a variety of sample types and showed its vast usability for a multitude of experiments. While we focused on the analysis of non-isotopically labeled samples from various organisms in the presented work, we have also utilized the produced SILISs successfully in our NAIL-MS studies in bacteria [26] and yeast [21]. 

We have studied the effect of nutrition on the abundance of modified nucleosides in the tRNA of *E. coli* by growing the bacteria in either rich LB broth or minimal M9 salt solution. Interestingly, we found only minimal to no effects of the medium on the abundance of the studied nucleosides (analysis in stationary phase). Furthermore, we analyzed the modification profile of *P. aeruginosa* tRNA. While the overall modification profile is similar to the profile of *E. coli*, three modified nucleosides were significantly different. The first is s^4^U, a modified nucleoside commonly located at position 8 of bacterial tRNAs. This modification is known to act as a sensor for UV irradiation [39], which leads to delayed growth during UV light exposure [40]. At this point, it is difficult to judge how the reduced abundance of s^4^U in *P. aeruginosa* affects its UV sensitivity. From the literature it is known, that *P. aeruginosa* has a higher UV-A sensitivity compared to *E. coli* [41], but it remains unclear how this finding is connected to the abundance of s^4^U. In addition to s^4^U, Gm is 4-fold lower in *P. aeruginosa* compared to *E. coli*. It would be interesting to compare the TLR7 activation of total tRNA from both organisms [30].

We have used the yeast tRNA SILIS to study the abundance of modified nucleosides in total tRNA from various eukaryotic organisms. We find the same chemical diversity of modified nucleosides in all analyzed tRNAs, however, *C. elegans* has a very low abundance of most modified nucleosides. Only the ribose-methylated nucleosides are of similar abundance compared to the other organisms and Am is significantly higher. Our analysis used tRNA prepared by size exclusion chromatography. It is possible, that worms have a unique set of ribose-methylated RNAs which are of comparable length to tRNA. It is possible that these RNAs dilute the actual tRNA modification profile and thus the low modification density can be explained. This unique modification profile of *C. elegans* and other worms should be subject to future studies. For the mammalian cells and tissue samples, we find comparable modification densities in the tRNAs. The comparison of HEK and mouse tissue tRNA showed a slightly lower abundance of structure stabilizing tRNA modifications such as 1-methyladenosine, 7-methylguanosine and 5-methyluridine in the tissue-derived tRNAs. The nutritional environment of cultured cells might be the explanation for these minor differences. Our results for mouse brain and mouse liver are comparable to previous reports [13]. In addition, we find a 2- and 1.5-fold reduced abundance of Gm and Um, respectively, in the mouse brain tRNA compared to the mouse liver tRNA. Gm is located at position 34 (wobble) of tRNA^Phe^_GAA_ and Um in tRNA^Glu^_UUG_ and thus both modifications are directly linked to translation. In fact, defects in tRNA anticodon 2′-*O*-methylation are implicated in nonsyndromic X-linked intellectual disability due to mutations in *FTSJ1* [42]. Therefore, we assume that the lower abundance of these modified nucleosides in brain tRNA is of physiological importance.

The analysis of eukaryotic hypermodified uridine derivatives, the wobble uridines, has always proven to be difficult. In our hands, the alkaline reaction conditions of the enzymatic digestion reaction (dictated by the mandatory use of alkaline phosphatase) results in a slow chemical decomposition of the modified nucleosides mcm^5^U, mcm^5^s^2^U and ncm^5^s^2^U. Storage of digested tRNA at −20 °C does not stop this decomposition. From our experience, the addition of 1/3 Vol of our LC–MS mobile buffer (pH 5.3) after the digestion slows down the decomposition process. Nevertheless, analysis should be performed on the same day. A SILIS for wobble uridine quantification cannot be stored either and should be produced on the same day of sample analysis. Therefore, we recommend keeping aliquots of 5–10 µg total tRNA (from SILIS production) and digest once needed. Higher quantities of s^2^U and ncm^5^s^2^U SILIS can be produced in knockout *S. c.* strains [43].

The bottleneck in wobble uridine analysis is the low abundance and detection efficiency of these modified nucleosides in tRNA. For mRNA modification analysis, the key bottleneck is the production of truly pure samples. We have used a stringent purification protocol, which uses a series of Poly-A enrichment, Ribominus treatment and again Poly-A enrichment steps to produce mRNA samples of best achievable purity. Such a procedure is necessary to ensure removal of especially ribosomal RNA as recently presented [35]. Although the procedure is highly expensive (~200 € per sample), the purity of the sample is not guaranteed. As we show in our chip electropherogram, the yield of total mRNA is very low and small amounts of tRNA and rRNA contaminations may be hidden within the mRNA signal. Our results reflect our circumspection. We detect a variety of modified nucleosides in our purified mRNA samples, but all in very low abundance. A validation to confirm their nature as true mRNA modifications is currently only possible for m^6^A, m^7^G, m^1^A, m^5^C and m^3^C by modern sequencing techniques. Due to the nature of LC–MS/MS analysis and the necessarily complete digestion, the biological meaning is difficult to interpret. Nevertheless, LC–MS/MS analysis, and especially absolute quantification using metabolically produced SILIS, remains a key tool to study the abundance and potential dynamics of modifications in mRNA once their presence has been verified using orthogonal techniques. 

## Figures and Tables

**Figure 1 genes-10-00026-f001:**
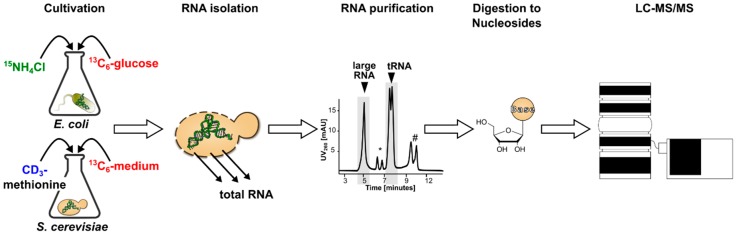
To produce stable isotope-labeled internal standards (SILIS), *Escherichia coli* (*E. coli*) or *Saccharomyces cerevisiae* (*S. cerevisiae*) were cultured in isotope labeled media using nutrients with the highest isotope-purity possible. The total RNA was isolated and then separated into transfer RNA (tRNA) or large/ribosomal RNA through size exclusion chromatography. (* indicates 5.8 and 5S rRNA and # phenol) The different RNAs were enzymatically digested into nucleosides and analyzed by liquid chromatography–mass spectrometry (LC–MS/MS).

**Figure 2 genes-10-00026-f002:**
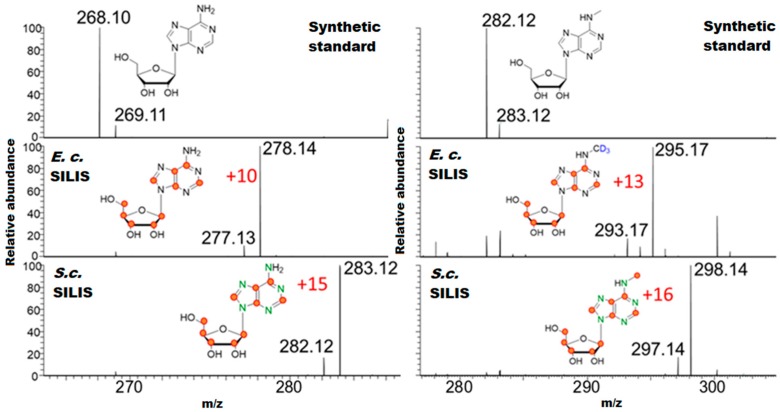
The mass spectra of adenosine (**left**) and 6-methyladenosine (m^6^A, (**right**)) are shown for the synthetic standard, the *E. coli* stable isotope-labeled internal standard (*E. c.* SILIS) and *S. cerevisiae* SILIS (*S. c.* SILIS). Atoms that were stable isotope labeled are indicated in red for carbon-13, blue for deuterium and green for nitrogen-15.

**Figure 3 genes-10-00026-f003:**
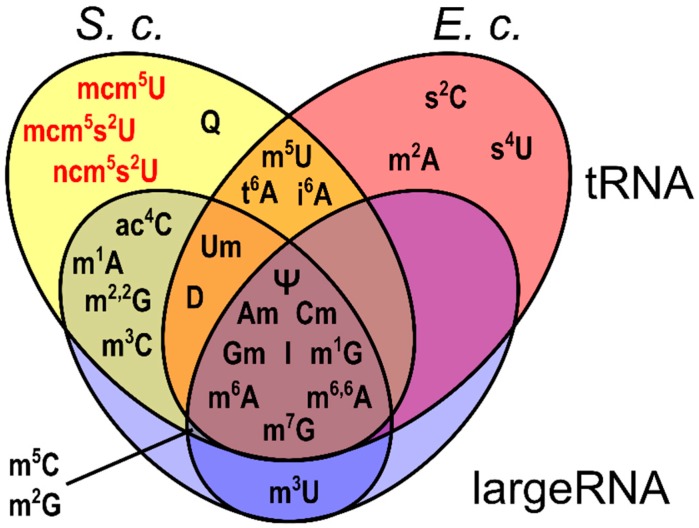
Venn diagram of modified nucleosides, which are produced in the presented SILIS preparation. *S. c.*: *Saccharomyces cerevisiae*; *E. c.*: *Escherichia coli*. Color code: tRNAs, red and yellow; large RNAs, blue. Red text: Wobble uridines. The full names of the abbreviated modified nucleosides are found in Table 1.

**Figure 4 genes-10-00026-f004:**
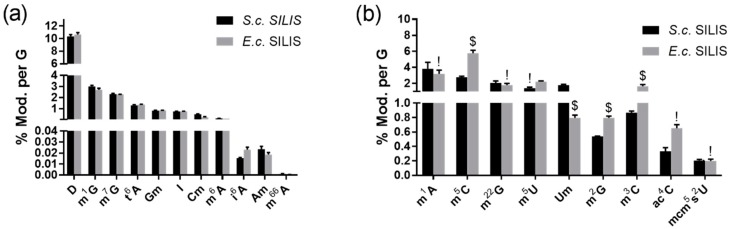
Quantification of modified (Mod.) nucleosides in tRNA from HEK 293T cells using either the *S. c.* SILIS or the *E. c.* SILIS. The average of three technical replicates is shown. The error bars reflect the standard deviation. (**a**) Modified nucleosides that are present in both SILISs. (**b**) Modified nucleosides, that are only present in the *S. c.* SILIS and not in the *E. c.* SILIS. For quantification of nucleosides absent in the *E. c.* SILIS, chemically similar SILIS analytes ($) or no SILIS analytes (!) (external calibration only) were used as detailed in the text.

**Figure 5 genes-10-00026-f005:**
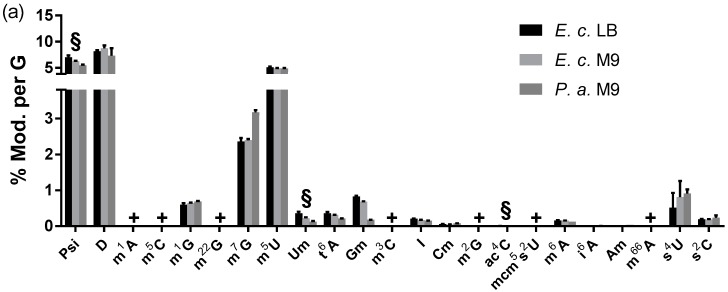
Modification profiles of tRNA from various organisms. (**a**) Comparison of *E. coli* (*E. c.*) tRNA grown in lysogeny broth (LB) or M9 medium and *P. aeruginosa* (*P. a.*) tRNA grown in M9 medium. §: for these modified nucleosides, the yeast SILIS was used. For the other bacterial modifications, the *E. c.* SILIS. (**b**) tRNA modifications in *S. cerevisiae* (*S. c.*), *Dictyostelium discoideum* (*D. d.*) and *Caenorhabditis elegans* (*C. e.*) (**c**) tRNA modifications in tRNA from HEK 293T cells, mouse liver (*M. m.* liver) and mouse brain (*M. m.* brain). #: These modified nucleosides are absent in the used yeast SILIS and thus s^4^U and s^2^C could not be quantified in these samples. + These modifications are below the LLOQ. The average of n = 3 biol. replicates are plotted. The exceptions are: HEK, *P. a.,* and *S. c.* each having one biol. replicate and three technical replicates. The error bars reflect the standard deviation.

**Figure 6 genes-10-00026-f006:**
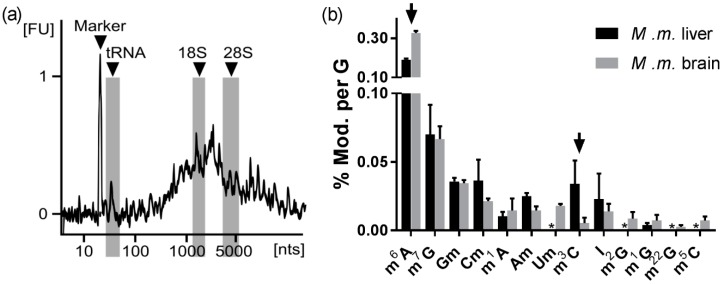
Absolute quantification of mRNA purified from mouse liver and mouse brain. (**a**) The chip electropherogram of mouse brain mRNA shows the expected size distribution from 500–7000 nucleotides. The location of tRNA and ribosomal RNA (18S and 28S) are indicated in grey. (**b**) Modified nucleosides found and quantified in mouse mRNA from brain and liver. Legend: Arrow: Most interesting differences between liver and brain tissue mRNA. *signal detectable but not quantifiable. The average of n = 3 biol. replicates are plotted. The error bars reflect the standard deviation.

**Table 1 genes-10-00026-t001:** Overview of modified nucleosides found in the respective RNA preparations from yeast (*S. cerevisiae*) and *E. coli*. Present: ✓; Absent: 0.

Modification	Yeast tRNA	Yeast Large RNA	*E. coli* tRNA	*E. coli* Large RNA
1-methyladenosine (m^1^A)	✓	✓	0	0
1-methylguanosine (m^1^G)	✓	✓	✓	✓
2-methyladenosine (m^2^A)	0	0	✓	0
2-thiocytidine (s^2^C)	0	0	✓	0
2′-*O*-methyladenosine (Am)	✓	✓	✓	✓
2′-*O*-methylcytidine (Cm)	✓	✓	✓	✓
2′-*O*-methylguanosine (Gm)	✓	✓	✓	✓
2′-*O*-methyluridine (Um)	✓	✓	✓^l^	0
3-methylcytidine (m^3^C)	✓	✓	0	0
3-methyluridine (m^3^U)	0	✓	0	✓ ^l^
4-thiouridine (s^4^U)	0	0	✓	0
5-carbamoylmethyl-2-thiouridine (ncm^5^s^2^U)	✓	0	0	0
5-methoxycarbonylmethyluridine (mcm^5^U)	✓	0	0	0
5-methoxycarbonylmethyl-2-thiouridine (mcm^5^s^2^U)	✓	0	0	0
5-methylcytidine (m^5^C)	✓	✓	0	✓
5-methyluridine (m^5^U)	✓	0	✓	0
7-methylguanosine (m^7^G)	✓	✓	✓	✓
N2-methylguanosine (m^2^G)	✓	✓	0	✓
N2,N2-dimethylguanosine (m^22^G)	✓	✓	0	0
N4-acetylcytidine (ac^4^C)	✓	✓	0	0
2-methylthioadenosine (ms^2^A)	✓	0	0	0
N6-isopentenyladenosine (i^6^A)	✓	0	✓	0
N6-methyladenosine (m^6^A)	✓	✓	✓	✓
N6,N6-dimethyladenosine (m^66^A)	✓	✓	✓	✓
N6-threonylcarbamoyladenosine (t^6^A)	✓	0	✓	0
Dihydrouridine (D)	✓	✓	✓	0
Inosine (I)	✓	✓	✓	✓
Pseudouridine (Psi or Ψ)	✓	✓	✓ ^2^	✓ ^2^
Queuosine (Q)	✓	0	0	0

^1^ Very low abundance in this RNA. If available, use yeast RNA instead. ^2^ Due to matrix effects, this modification cannot be reliably quantified in our chromatographic system.

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
