# Peer review of "Production and Application of Stable Isotope-Labeled Internal Standards for RNA Modification Analysis"

_genes, 2019, doi:10.3390/genes10010026_

Reviewer 1 Report

The manuscript describes the production of stable isotope labeled internal standards for potential application in RNA modification analysis. The authors used E. coli and S. cerevisiae as vehicular platforms to make the labeled modified ribonucleosides. By using the calibration curves of synthetic standards, they estimated the lower limits of quantification of isotope-labeled modified nucleosides through a dilution series. Using this as a benchmark they estimated the quantities of modified nucleosides in various sample types. They used MRM based detection of the m/z values of ribonucleoside and nitrogenous base through a triple-quad mass spectrometer.Subsequently, they conducted the RNA modification analysis in different organisms and tissue types.

Novelty of the new method:

Lines 278-279: The quantification requires synthetic standard to calculate response factor or nucleoside isotope factor. The modified nucleoside levels in biological systems are highly variable depending on the growth stage, culture conditions and cannot be quantified without the availability of synthetic standards. Production of isotopomers and quantification based on its use were previously documented by the lead author. It would be helpful with better explanation of how the current work is different from the  previously published work by the lead author.

 Lines 350-354: Not clear, how the non-optimal ionization and collision parameters can distinguish sample and SILIS canonical nucleoside signal and quantify the canonical nucleosides and the modified nucleosides within a single sample run.

They report the presence of a number of modified nucleosides in mRNA which are known to be present in tRNA and rRNA. The authors state that the amount of isolated mRNA is rather low, and they cannot completely exclude contamination by remnant tRNA or rRNA hidden underneath the mRNA population (lines 488-490), yet they report a number of nucleoside modifications that have been documented in tRNA and rRNA. The author should make a qualifying statement saying that the existence of such modifications in mRNA should further be verified by orthogonal methods before making a general conclusion.

Author Response

Please find the response attached below

Reviewer 2 Report

This manuscript describes a method for absolute quantitative analysis of the modified nucleosides obtained following complete digestion of an RNA of interest. Using this method, the fractions containing tRNA and mRNA molecules isolated from multiple prokaryotic and eukaryotic cells, organisms, and tissues, were shown to have distinct modification profiles, and the abundances of the individual modified residues spanning 2 orders of magnitude.  The scientific content of the paper is solid and unique observations using a novel quantitative MS tool are provided, however the presentation of the material, including the title/ abstract, parts of the text and plots, would benefit from modifiction.

Major concern:

The focus of the paper may not be fully appropriate for “GENES”. As indicated in the title and the abstract, the major goal is to provide a protocol for preparation of the internal standard usable for quantitative analysis of individual modifications. The authors have extensively shown the applicability of their method, which is their major strength, however biological implications of their measurements remain beyond the scope of the study.  The material presented might have higher impact if focus is switched from methodology to application, and more description provided to emphasize the role of the quantified modifications in regulating gene expression and in human health. 

--just a brief note from the editor handling the manuscript: 'Technologies' section stands for new methodologies/tools/tests/software, etc, so the current author contribution does fall into the scope of the Genes journal ---

Moderate concerns:

1.     The work described in this manuscript offers an improved method for absolute quantitation of the modified nucleosides using SILISs isolated from two different organisms ( E.coli and S. cerevisiae) and  thus increasing the coverage from 15 to 25+ analytes. Since prior work from this lab (Kellner et al., NAR,2014, where SILIS was first introduced)  lay the foundation for quantitative analysis, it would be useful to include a paragraph  describing  strengths and weaknesses of the existing method, the current needs in the field of RNA modification analysis , and how they were addressed at present.
 A general concern is that the flow of the paper is interrupted, making it difficult for the reader to grasp the critically important components and steps of the method.  We suggest that lines 217-227  be revised, and to include Figure S1 or its alternative in the main text, providing more clarity to the whole workflow and not just the initial steps of the SILIS preparation.    

2.     The choice of the metabolic precursors for S. cerevisiae SILIS should be further explained. The authors mention (lines 71-73) that deuterium labeled precursors are not ideal, as they introduce a RT shift with respect to the unlabeled analyte of interest,  and we agree that this may lead to somewhat inaccurate quantitation. Here, CD3-methionine was chosen in combination with 13C-glucose for metabolic labeling of RNA in yeast, resulting in complex looking spectra due to incomplete 13C labeling. Why, 13C-glucose alone was insufficient for internal standard preparation? Were the forementioned RT shifts affecting quantitative analysis using yeast SILIS?

3.  The need for separating tRNAs from large RNAs standards should be better defined. The nucleoside composition of both isotopically labeled fractions have been characterized in details ( section 3.1.1), but only tRNA SILISs were used for quantitation, including analysis of the mRNA modifications  found in mouse tissues.  The paper would strongly benefit from describing the possible uses of the large RNA SILISs, as well as the requirements for a SILIS mixture in general (e.g, composition, dynamic range and concentrations of the modified nucleosides, concentrations of the canonical nucleosides etc.)  

4.  Here, abundance of a modified nucleosides is reported as a fraction of guanosines present in the sample, to normalize to the amount of RNA biomass.  The strong concern is that the measurements thus become dependent on the purity of the sample. For example, tRNA samples from different eukaryotic organisms used in this work may be contaminated with RNA species of similar molecular weight (5.8S, 5S, snoRNA, snRNA) , and their contribution may strongly depend on the origin. The contaminating species will unequally contribute to the pools of the modified residues and the guanosine pool, overall skewing the measurements. 

5.  We suggest that the plots Figures 4 and 5 be presented in a log scale. This will help to simultaneously see the high and low abundance values and associated error bar. The break of the Y axis should be removed as it obscures the readings. Importantly, authors should describe the origin of the reported measurement errors.

For example, half of the values reported  in the Figure 5a plot remain unreadable due to a 100 fold difference in the  relative abundance of the bacterial tRNA modifications. 

6.  Figure 4a plot was included in the text to demonstrate that nucleoside quantitation is independent on the origin of the internal standard mixture. However, values for a few residues (I, Cm, Gm, and possibly t6A) obtained using  bacterial vs yeast SILIS seem to vary above the measured error. Please, explain the observed variations.  In the 4b plot, the use of the E. coli SILIS in the figure legend is confusing, since the residues shown are absent in the E. coli tRNAs. The authors should color code the measurements obtained using chemically similar analytes and external calibrations, which may reinforce the need of the internal standards for accurate and reproducible quantitation.

Minor corrections:

1.     Line 70 : I believe, “isotopomers” should be changed to “isotopologues”, since the light and heavy labeled species have different isotopic composition.

2.     In Material and Methods, please, clarify why LC-MS methods described in sections 2.9 and 2.11 were using slightly different HPLC conditions.

3.     In Material and Methods, section 2.8: The authors should clarify whether digestion reaction mixture was used directly for LC-MS sample/SILIS preparation without prior removal of the enzymes.

4.     Not all supplemental material if referenced in the main text: e.g., Tables S3, S4 references are not used. Additionally, it is likely that Tables S3 contain redundant information (already included in S2) and may be removed.

5.     Figure 3 can be moved to the Supplemental Material, since Table 1 includes the list of modifications present in each of the four samples. Color coding of the of the Venn sectors seems excessive and unused.

Author Response

Please find the response attached below

Round  2

Reviewer 1 Report

The authors did well to address the concerns in the revised manuscript. Some suggestions to improve the clarity are as follows:

The figure legends are currently abbreviated. Since the figure size is large, the legends could be expanded for clarity.  This is especially true for the legends specifying the scientific names.